# On the Generalization Ability of Data-Driven Models in the Problem of Total Cloud Cover Retrieval

**Mikhail Krinitskiy** *[ID], **Marina Aleksandrova, Polina Verezemskaya** [ID], **Sergey Gulev, Alexey Sinitsyn, Nadezhda Kovaleva and Alexander Gavrikov**

Shirshov Institute of Oceanology, Russian Academy of Sciences, Moscow 117997, Russia; marina@sail.msk.ru (M.A.); verezem@sail.msk.ru (P.V.); gul@sail.msk.ru (S.G.); sinitsyn@sail.msk.ru (A.S.); nadia@sail.msk.ru (N.K.); gavr@sail.msk.ru (A.G.)
* Correspondence: krinitsky@sail.msk.ru; Tel.: +7-499-1247928

**Abstract:** Total Cloud Cover (TCC) retrieval from ground-based optical imagery is a problem that has been tackled by several generations of researchers. The number of human-designed algorithms for the estimation of TCC grows every year. However, there has been no considerable progress in terms of quality, mostly due to the lack of systematic approach to the design of the algorithms, to the assessment of their generalization ability, and to the assessment of the TCC retrieval quality. In this study, we discuss the optimization nature of data-driven schemes for TCC retrieval. In order to compare the algorithms, we propose a framework for the assessment of the algorithms' characteristics. We present several new algorithms that are based on deep learning techniques: A model for outliers filtering, and a few models for TCC retrieval from all-sky imagery. For training and assessment of data-driven algorithms of this study, we present the Dataset of All-Sky Imagery over the Ocean (DASIO) containing over one million all-sky optical images of the visible sky dome taken in various regions of the world ocean. The research campaigns that contributed to the DASIO collection took place in the Atlantic ocean, the Indian ocean, the Red and Mediterranean seas, and the Arctic ocean. Optical imagery collected during these missions are accompanied by standard meteorological observations of cloudiness characteristics made by experienced observers. We assess the generalization ability of the presented models in several scenarios that differ in terms of the regions selected for the train and test subsets. As a result, we demonstrate that our models based on convolutional neural networks deliver a superior quality compared to all previously published approaches. As a key result, we demonstrate a considerable drop in the ability to generalize the training data in the case of a strong covariate shift between the training and test subsets of imagery which may occur in the case of region-aware subsampling.

**Keywords:** total cloud cover; all-sky camera; algorithms assessment; neural networks; machine learning; data-driven approach

## 1. Introduction

It is generally considered that clouds play one of the primary roles in climate by mediating short wave and long wave radiative fluxes [1–3]. Clouds are also crucial for the hydrological cycle on both a global and regional scale [4]. Cloud cover plays a vital role in regulating climatic feedback and thus cloud cover may be exploited as a diagnostic in sensitivity studies of climate models in different scenarios. Cloud cover variability over the ocean is a key variable for understanding global and regional circulation phenomena and modes of variability, such as monsoons, ENSO (El Niño Southern Oscillation), Intertropical Convergence Zone (ITCZ) shift, North Atlantic Oscillation (NAO), and Pacific Decadal Oscillation (PDO). Clouds also strongly impact sea-air interactions in boundary currents zones and upwelling zones [5,6].

Clouds also play a major role in various processes, like solar power production and air traffic dispatching. In scientific observations, clouds may be unavoidable obstacles.

In satellite observations, clouds mask land and ocean surface, thus significantly decreasing the rate of useful satellite information about the surface in cloud-rich regions. In optical observations in land-based astronomy, clouds are unavoidable obstacles as well [7,8]. Thus, accurate estimating and forecasting of cloud characteristics is crucial for observational time planning [9,10].

There are a number of data sources available for studies of clouds in the ocean. Among the most frequently used are remote sensing archives, data from reanalyses, and observations made at sea from research vessels and voluntary observing ships. Each of these data sources have their own advantages and flaws. Satellite observations may be considered accurate, and they are uniformly scattered spatially and temporally, though their time series are limited as they started only in the early 1980s [11–16]. Satellite measurements are also characterized by different flaws, e.g., underestimating cloudiness over sea ice in nighttime conditions [17] which may be significant in the Arctic. Data from reanalyses is uniformly sampled as well. However, although the models applied in reanalyses for diagnostic cloud cover estimation continuously improve, they need further development and validation [17,18]. Reanalyses were shown to underestimate total cloud cover compared to measurements provided by land-based weather stations, and observations over the ocean [17,19]. A possible cause of this underestimation may be the overestimated downward short-wave radiation that is taken into account within the computations for cloud coverage [20]. It is worth mentioning, however, that in some particular cases reanalyses may be consistent with meteorological stations data in terms of the low-frequency temporal variability of cloud characteristics [21].

The best data source for climatological studies of clouds is the archive of observations made from Voluntary Observing Ships (VOS) which are organized into the International Comprehensive Ocean-Atmosphere Data Set (ICOADS) [22,23]. The very first visual observations at sea were made in the middle of the nineteenth century, though there were not many before the twentieth century. Most studies use the ICOADS observations dated from the early 1950s [24–26]. The change of cloudiness codes in the late 1940s [27] reduces the validity of climatic studies relying on long-term homogeneity of the time series of cloudiness characteristics over the ocean in the twentieth century. The key disadvantage of the ICOADS records is their temporal and spatial inhomogeneity. Most observations are made along the major sea traffic routes in the North Atlantic and North Pacific. In contrast, the central regions of the Atlantic and Pacific are not covered by measurements tightly enough. The Southern ocean coverage is poor as well [28].

Visual observations of clouds are considered the most reliable at the moment [29,30]. The observations over the ocean are conducted every three or every six hours at UTC time divisible by 3 h. This procedure provides four or eight measurements a day per observing ship. Observed parameters include Total Cloud Cover (TCC) and low cloud cover, the morphological characteristics of clouds, and an estimate of cloud-base height. The total cloud coverage is estimated by a meteorology expert based on the visible hemisphere of the sky. To estimate the total cloud cover, the expert considers the temporal characteristics of the clouds being observed along with their additional parameters, e.g., precipitation, preceding types of clouds, light scattering phenomena, etc. For TCC retrieval, the observer visually estimates the fraction of the sky dome occupied by clouds. This procedure is described in detail in the WMO manual on codes [29], the WMO guide on meteorological observations [31], and in the International Cloud Atlas [30]. The procedure for estimating TCC involves making a decision on whether to consider the gaps in the clouds through which the sky is visible as "sky" or as "cloud". The decision is based on the cloud type. For example, the gaps are countered as "clear sky" in the case of low clouds or convective clouds, e.g., cumulus and stratocumulus clouds. In contrast, the sky gaps are not countered as "sky" in the case of cirrus, cirrocumulus, and almost all sub-types of altocumulus clouds. This feature of the TCC estimation procedure introduces uncertainty into the results of automated TCC retrieval schemes.

Cloud characteristics remain one of the few meteorological parameter subsets still observed by experts visually whereas most of the other indices are measured automatically today. The procedure is hard to automate due to the high amount of non-formalized rules of thumb and heuristics that are learned by an expert as a result of long-term practice. This way an expert estimating cloud characteristics adjusts their understanding of cloud formation processes, relates them to the state of clouds theory, and learns how the observed visuals correspond to the underlying physics dictated by the theory. The whole expert experience results in a somewhat consistent measurement of quantitative characteristics that are recommended by WMO as the most reliable source of information about clouds. The flaws of the approach of visual estimation are: It suffers from subjectivity; the learning curve mentioned above may result in biased estimates; and the approach itself is highly time-consuming and requires massive human resources even today, in the era of Artificial Intelligence and advanced computer vision.

In this study, we discuss mostly the problem of data-driven TCC retrieval, although the classification of observed clouds is also an intriguing problem addressed in a number of studies employing data-driven methods along with expert-designed and fused approaches [32–37].

### 1.1. On the Optimization Nature of the Known Schemes for TCC Retrieval from All-Sky Optical Imagery

A number of automated schemes were proposed in last 20 years, beginning with the pioneering work of Long et al. presented along with the optical package for the all-sky imagery retrieval in 1998 [38] and described in detail in 2006 [39]. Since the first scheme of Long et al., numerous variations of algorithms were described for estimating some quantitative characteristics based on different in situ measurements such as all-sky imagery [38,40–45] or downward short-wave radiation [46]. Most of these algorithms are designed by experts integrating their own understanding of the physical processes that result in all-sky imagery similar to the ones presented in Figure 1a,b.

Given all-sky imagery acquired, in most simple cases, an index is calculated pixel-wise, e.g., red-to-blue ratio (RBR) in a series of papers by Long et al. [38,39,47] or in the following studies [48–50], or the ratio $\frac{B-R}{B+R}$ in [42,43,51], or even a set of indices [52]. Then, an empirical threshold is applied for the classification of pixels into two classes: "cloud" and "clear sky". A few schemes with a more complex algorithm structure were presented recently [41,45,47,53,54]. These schemes were introduced mostly for tackling the flaw of simple yet computationally efficient schemes: Taking the sun disk and the circumsolar region of an all-sky image as "cloud".

In contrast with the expert-designed algorithms for TCC retrieval, only a few data-driven schemes have been presented lately for estimating TCC [55–57] or for clouds segmentation in optical all-sky imagery [58]. Researchers may consider the problem of TCC retrieval to be solved or to be too simple to address with complex machine learning algorithms. However, none of the presented approaches demonstrated any significant improvement in the quality of TCC estimation.

The only exception here is the method presented by Krinitskiy [57] which claimed to achieve an almost human-like accuracy in TCC estimation. There is, however, an error at the validation stage resulting in incorrect quality assessment. This study may be considered a corrigendum to the conference paper of Krinitskiy [57].

At this point we need to note, that the so-called data-driven methods for the approximation of some variable (say, TCC) do not differ considerably from the ones that are designed by an expert. An expert-designed method implies an understanding of the underlying processes that form the source data (say, optical ground-based imagery) and its features (say, relations between red, green, and blue channels of a pixel registering clear sky or a part of a cloud). Then, these human-engineered features are used in some sort of an algorithm for the computation of an index or multiple indices, which then are aggregated over the image to a quantitative measure. The algorithm may be considered simple [38,39,42,44,54], adaptive [45,53], it may be designed to be complex to some extent

in an attempt to take some advanced spatial features into account [40,41,47,49,50], or in an attempt to correct the distortion of imagery or other features of imagery that are not consistent with the initial researcher's assumptions [42,49,50,52]. However, all these methods rely on the aggregation step at some point that is commonly implemented as a variation of thresholding. The threshold value(s) is empirical and should be adjusted to minimize the error of TCC estimates or maximize the quality of the method that is proposed in the corresponding study. This adjustment stage is commonly described vaguely [49] or briefly [39], though in this sense, the above mentioned expert-designed algorithms are essentially data-driven and inherently have an optimization nature, thus the optimization is a key part of the development of these schemes.

In case of the schemes employing Machine Learning (ML) methods [55–57], the algorithms are inherently of an optimization nature since the essence of almost any supervised ML algorithm is the optimization of empirical costs based on a training data set.

In the context of this study, the mentioned training dataset consists of ground-based all-sky imagery with corresponding TCC estimates ("labels" hereafter). These estimates were made by an expert in the field at the same time as the image was taken. It is worth mentioning that this labeling procedure is subject to noise. There are multiple sources of noise and uncertainties in labels and the imagery itself:

- Subjectivity of an observer. As mentioned above, the whole prior expert experience may impact the quality of TCC estimates. The uncertainty introduced by a human observer has not been assessed thoroughly yet. We can only hope that this uncertainty is less than 1 okta (one eighth of the whole sky dome, the unit of TCC dictated by WMO [29,30]), though from the subjective experience of the authors, the uncertainty may exceed 1 okta when one scene is observed by multiple experts;

- Violation of the observations procedure. Ideally, an expert needs to observe the sky dome in an environment clear from obstacles, which may not always be the case not only in strong storm conditions at sea, but even in the case of land-based meteorological stations. In our study, every record made in hard-to-observe conditions was flagged accordingly, thus no such record is used in the filtered training, validation, and test datasets;

- Temporal discrepancy $\Delta t, s$ between the time of an observation and the time of acquisition of corresponding imagery. This time gap can never be zero, thus a decision always needs to be made on whether $\Delta t$ is small enough to be considered negligible, and the expert records to be considered correct for the corresponding all-sky image;

- Reduced quality of imagery. There is always room for improvement in the resolution of optical cameras, their light sensitivity, and corresponding signal-to-noise ratio in low-light cases (e.g., for registering in nighttime conditions). The conditions of imagery acquisition may play a role as well, since raindrops, dust, and dirt may distort the picture significantly and may be considered strong noise in source data;

- Reduced relevance of the acquired imagery to the TCC estimation problem. It should be mentioned that in storm and high waves conditions in the ocean, an optical package tightly mounted to the ship may partly register the sea surface instead of the sky dome. In our experience, the inclination may reach $15°$, and even $25°$ in storm conditions. This issue, along with the uncertain heading of the ship, also prevents the computation of the exact sun disk position in an all-sky image;

- Reduced relevance of the acquired imagery to the labeling records. Ideally, the TCC labeling site should be collocated with the optical package performing the imagery acquisition. This is not exactly the case in some studies [41].

All these factors may be considered as introducing noise to the labels or imagery of the datasets that are essentially the basis of all the data-driven algorithms mentioned above. The impact of these factors may be reduced by modifying the observation and imagery acquisition procedures, or by data filtering. Some factors may be addressed by the releases of more strictly standardized procedures for cloud observations, though the WMO guide seems strict and straightforward enough [29,30].

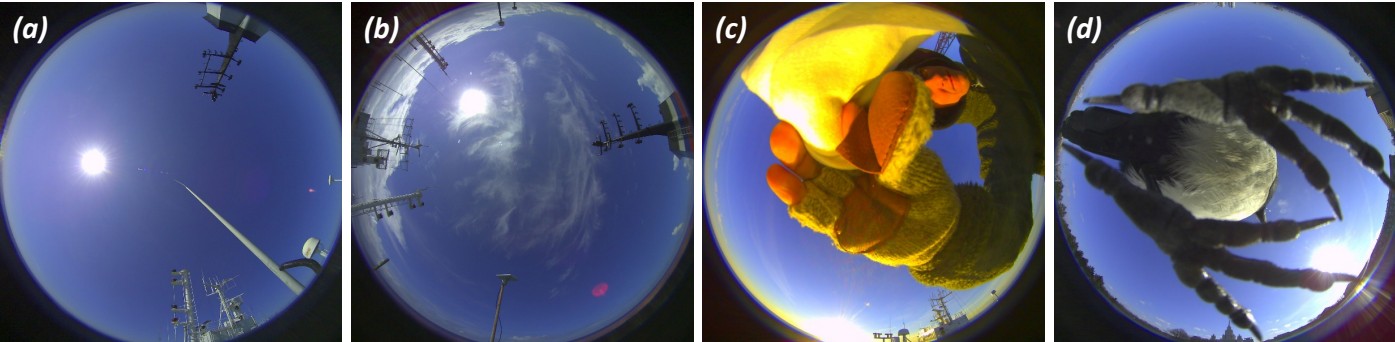

**Figure 1.** Examples of all-sky images acquired by optical package SAILCOP (Sea-Air Interactions Laboratory Clouds Optical Package) (see Figure 2): (**a**,**b**) Typical all-sky imagery over the ocean; (**c**,**d**) rare cases (outliers) when an image represents a scene with a bird or maintenance staff.

However, there are factors that are unavoidable, that were not mentioned above. Typical types of clouds and their amounts differ in various regions of the ocean. Typical states of the atmosphere and its optical depth differ significantly as well, strongly influencing the quality of imagery and its features. Statistically speaking, for the development of a perfect unbiased TCC estimator within the optimization approach, one needs to acquire a training dataset perfectly and exhaustively representing all the cases and conditions that are expected to be met at the inference phase. This requirement seems unattainable in practice. Alternatively, a data-driven algorithm is expected to have some level of generalization ability, thus being capable of inferring TCC for new images acquired in previously unmet conditions. The effect of significant changes in the source data is well known in machine learning and in the theory of statistical inference as covariate shift. The capability of an algorithm to generalize is called generalization ability. This ability may be expressed in terms of discrepancy between the quality of the algorithm assessed on different datasets within one downstream task.

### 1.2. On the Climatology of Clouds

Climatology of clouds shows a big difference between characteristics of clouds in various regions of the world ocean not only in terms of total cloud cover, but in terms of typical cloud types and their seasonal variability. In Aleksandrova et al. 2018 [28], the climatology of total cloud coverage for the period of 1950–2011 is presented.

The climatology is based on ICOADS [22,23] data and limited to periods from January to March, and from July to September. In the tropics, the average TCC varies from 1.5 to 3 okta, whereas the average TCC in the middle latitudes and sub-polar regions ranges from 6.5 to 7.5 okta. In the middle latitudes and sub-polar regions of the ocean, seasonal variability results in an increase in TCC in summer compared to winter. In contrast, in the tropics and subtropics, the average TCC in summer is lower than in winter, which is especially noticeable in the Atlantic ocean. The most striking seasonal variations are registered in the Indian ocean, which is characterized by strong monsoon circulation. That is, in the northern regions of the Indian ocean, in winter dry seasons, the mean seasonal TCC varies from 1 to 3 okta, whereas in wet summer seasons the average seasonal TCC is from 5 to 6.5 okta. A time series of the total cloud coverage for distinct regions of the world ocean demonstrate significant differences in characteristics of the inter-annual variability of TCC. Cloud coverage in the southern Atlantic rarely drops below 80–90%, whereas the inter-annual seasonally averaged TCC variability may exceed 50%. In the central Pacific, some years may be considered outliers with TCC significantly exceeding the mean values due to ENSO [26].

At the same time, averaged TCC values are not representative enough within the scope of our study. One needs to consider the regimes of cloudiness over the regions of the ocean. In the middle latitudes, both the regime of long-lasting broken clouds (4–6 okta), and the regime of interchanging short periods of overcast and clear-sky conditions may

result in the same seasonal mean TCC. The diversity of the regimes of cloudiness may be shown by the empirical histograms of the fractional TCC for various regions of the world ocean (see [28], Figure 5).

Types of clouds are not distributed evenly over the ocean as well. This feature strongly impacts the average TCC as well as the characteristics of the acquired imagery. The frequency of different types of clouds varies between the tropics, mid-latitudes, and sub-polar regions. The difference between the western and eastern regions of oceans is significant as well, especially when one considers low clouds [59].

Cumulonimbus, denoted by two codes in the WMO manual on codes ($C_L$9 for *cumulonimbus calvus* and $C_L$3 for *cumulonimbus capillatus*) [29] are observed frequently in the tropics, are rare in the mid-latitudes, and can almost never be registered in sub-polar regions of the world ocean. At the same time, there is a considerable difference even between the distributions of the two sub-types of cumulonimbus over the ocean: Records of *cumulonimbus calvus* over the ocean are twice as frequent as those of *cumulonimbus capillatus*. There is also a significant difference in spatial distributions of these two sub-types: *Cumulonimbus capillatus* are registered sometimes in the North Atlantic and northern regions of the Pacific ocean, whereas *cumulonimbus calvus* are observed almost only in the tropics and the equatorial zone. *Cumulonimbus calvus* has a maximum frequency in the central regions of the oceans, whereas *cumulonimbus capillatus* are more frequent in coastal zones.

*Cumulus* clouds include two WMO codes: $C_L$1 for *cumulus humilis* and $C_L$2 for *cumulus mediocris* or *cumulus congestus*. The frequency of *cumulus* clouds is high in the western and central regions of the subtropics and tropics of the oceans. In the mid-latitudes, *cumulus* clouds are not that frequent in general, and even less frequent in summer. There are however exceptions, which are the eastern part of the North Atlantic and the northern regions of the Pacific ocean, where cold-air outbreaks are more frequent, thus the conditions for *cumulus* clouds are more favorable. Generally, *cumulus humilis* are less frequent over the ocean compared to *cumulus mediocris* and *cumulus congestus*.

In contrast with *cumulus* clouds, *stratus* clouds are much more frequent in mid-latitudes (WMO codes $C_L$5 for *stratocumulus* other than *stratocumulus cumulogenitus*, and $C_L$6 for *stratus nebulosus* or *stratus fractus* other then *stratus* of bad weather). They are also frequent in eastern regions of the subtropics over the ocean. In some studies, these two codes are considered as one type [60,61], however, their spatial distributions differ considerably. *Stratocumulus* clouds ($C_L$5) are registered most frequently in eastern regions of the oceans' subtropical zones, whereas *stratus* clouds ($C_L$6) are mostly found in the mid-latitudes, especially in summer. *Stratus* clouds ($C_L$6) are rare in eastern regions of the subtropics (excluding some of the upwelling zones). There are also *stratus fractus* or *cumulus fractus* of bad weather (WMO code $C_L$7), which are frequently observed in the mid-latitudes in winter, when the synoptic activity is strong. Sometimes, clouds of $C_L$7 are registered in the low latitudes, in the stratiform precipitation regions [62].

Sometimes there are even no low clouds (WMO code $C_L$0). This code is frequently registered in the coastal region of the ocean, in the Arctic, and the Mediterranean sea.

As one may notice from the brief and incomplete climatology of clouds above, different types of clouds are distributed very unevenly over the world ocean. If one collects a dataset in a limited number of the regions of the ocean for the optimization of a data-driven algorithm, the resulting scheme may lack quality in the regions that were not represented in the training set.

### 1.3. On Data-Driven Algorithms for TCC Retrieval from All-Sky Optical Imagery

A few data-driven methods were presented recently for estimating TCC from all-sky optical imagery [55,56,58]. In [56], the only improvement compared to simple schemes [39] is the application of a clustering algorithm in the form of a superpixel segmentation step. This step allows the authors to transform the scheme to an adaptive one. However, one still needs to compute the threshold value for each superpixel. In [58], a probabilistic approach for cloud segmentation is proposed employing Principal Components Analysis

(PCA) approach along with the Partial Least Squares (PLS) model. The whole approach may be expressed as PLS-based supervised feature engineering resulting in the pixel-wise linearly computed index claimed to be characterizing the probabilistic indication of the "belongingness" of a pixel to a specific class (i.e., cloud or sky). For this index to be technically interpreted as a measure of probability, it is normalized to the $[0, 1]$ range linearly. The model described in this study is similar to logistic regression with the only reservation being that the log-regression model has strong probabilistic foundations resulting in both the logistic function and binary cross-entropy loss function. The logistic function naturally transforms the covariates to the probability estimates within the $[0, 1]$ range without any normalization. Thus, the model proposed in [58] has questionable probabilistic foundations compared to well-known logistic regression. However, the study [58] is remarkable, as it is the first (to the best of our knowledge) to formulate the problem of cloud cover retrieval as a pixel-wise semantic segmentation employing a simple ML method. In [55], the authors employ the stat- of-the-art (at the time of the study) neural architecture namely U-net [63] for semantic segmentation of clouds in optical all-sky imagery. The two latter approaches are very promising if one has a segmentation mask as supervision. It is worth mentioning that labeling all-sky images in order to create a cloud mask is very time-consuming. In our experience, this kind of labeling of one image may take an expert 15 to 30 min depending on the amount of clouds and their spatial distribution. To the best of our knowledge, no ML-based algorithms were presented that are capable of estimating TCC directly without preceding costly segmentation labeling.

One more issue with most of the presented schemes for TCC estimation is the lack of universal quality measure. In some studies, the quality measure is not even introduced [52]. Other studies with the problem formulated as a semantic segmentation of clouds, employ typical pixel-wise quality measures adopted from computer vision segmentation tasks, such as Precision, Recall, F1-score, and misclassification rate [56,58]. This decision may be motivated by the models applied and by the state of the computer vision. However, the definition of a quality measure should never depend on the way the problem is solved. In some studies, the quality measures that are used are common for regression problems, e.g., correlation coefficient [55], MSE (Root Mean Squared Error), or RMSE (Root Mean Squared Error) [49]. In probability theory, these measures usually imply specific assumptions about the distribution of the target value (TCC), and also assumptions about the set of all possible outcomes and their type (real values). In the case of TCC, these assumptions are obviously not met. It is also obvious that the assessment of the quality of TCC retrieval by any valid algorithm (that does not produce invalid TCC) is biased for the events labeled as 8 okta or 0 okta. Since the set of possible outcomes of TCC is limited, any non-perfect algorithm underestimates TCC for the 8-okta events and overestimates TCC for the 0-okta events. Thus, any quality metric is biased by design as it is calculated using the deviation of the result of an algorithm from the expert label. We are confident that one should never use a biased-by-design quality measure. Thus, if one were to employ the quality measures of regression problems (MSE, RMSE, correlation coefficient, determination coefficient, etc.), it would be consistent with solving the problem as regression, which is not always the case for the studies mentioned above.

In our understanding, the problem of TCC retrieval should be formulated as a classification since the set of possible outcomes is finite and discreet. Alternatively, one may formulate the problem as ordinal regression [64]. In these cases, accuracy (event-wise, rather than pixel-wise) or other quality measures for the classification problem may be the right choice. In our study, we balance the datasets prior to the training and quality assessment, thus accuracy may be considered a suitable metric. In the case of ordinal regression, the categorical scale of classes is implied, which shows an order between the classes. This is exactly the case in the problem of TCC retrieval, since the classes of TCC are ordered in such a way that the label "1 okta" denotes more clouds compared to the label "0 okta"; "2 okta" is more than "1 okta", and so on. In this case, the conditional distribution of the target variable $P(TCC|event)$ is still not defined, which would be necessary for the formulation of

a loss function and quality measures (MSE, RMSE, etc.) within the approaches of Maximum Likelihood Estimator or Maximum a Posteriori Probability Estimator, similar to regression statistical models. However, the "less or equal than one-okta error accuracy" ("Leq1A" hereafter) is frequently considered as an additional quality measure in the problem of TCC retrieval [49,55]. In our understanding, this metric is still not valid and may be biased due to the reasons given above, though we include its estimates for our results in order to be comparable to other studies.

In the context of the introduction given above, the contributions of our study are the following:

1. We present the framework for the assessment of the algorithms for TCC retrieval from all-sky optical imagery along with the results of our models;
2. We present a novel scheme for estimating TCC over the ocean from all-sky imagery employing the model of convolutional neural networks within two problem formulations: Classification and ordinal regression;
3. We demonstrate the degradation of the quality of data-driven models in the case of a strong covariate shift.

The rest of the paper is organized as follows: In Section 2, we describe our Dataset of All-Sky Imagery Over the Ocean (DASIO); in Section 3.1, we describe the neural models we propose in this study and the design of the experiment for the assessment of their generalization ability; and in Section 4, we present the results of the experiment. Section 5 summarizes the paper and presents an outlook for further study.

## 2. Data

### 2.1. Dataset of All-Sky Imagery Over the Ocean (DASIO)

Since the early 2000s, we have been collecting all-sky imagery over the ocean along with concurrent expert estimates of the set of meteorological parameters recommended by WMO. Our experts observe and register TCC, low cloud cover, and other cloud characteristics among other parameters. From 2014 onwards, imagery acquisition was automated using the optical package designed and assembled in our laboratory [57]. We name it SAILCOP, which stands for "Sea-Air Interactions Laboratory Clouds Optical Package". SAILCOP is capable of acquiring the optical imagery of the visible skydome. In Figure 2, one registering head of the package is presented along with the mounting points of both of the optical heads on our research vessel. In Figure 1, some examples of the imagery are presented. One may notice that the positions of the superstructures of the vessel visible in the images in Figure 1 are not the same due to variations of the mounting points of cameras and variations of the host vessel itself. However, the mounting points are fixed for every mission. Thus, if one wants to apply the masking, the two masks (one per camera) are unique for a mission. The optical heads of SAILCOP are wired to the management computer. Each head is equipped with a GPS sensor and positioning sensors, including an accelerometer. As a result, SAILCOP is capable of taking pictures at moments when the position of the vessel is nearly horizontal. This capability helps us reduce the impact of the noise-introducing factor related to the swinging platform mentioned in Section 1.1. In addition, each pair of all-sky images is labeled with the GPS coordinates, date and time (UTC), and additional technical information. These attributes allow us to automatically compute the sun elevation in the ship's geographical position at the time of observations. Imagery acquisition with SAILCOP starts at dawn, when the sun elevation exceeds 0°, and stops when the sun elevation drops below 0°. In a regular functioning regime, SAILCOP takes images synchronously from two heads with a time discrepancy not exceeding 15 milliseconds. In Figure 3c,d, we present an example of the images acquired simultaneously from two cameras. One may see that they are almost the same. However, the two cameras were positioned some distance apart thus, the clouds are presented in these snapshots from slightly different angles. In some studies, this effect is exploited in the schemes for estimating cloud base height [65,66]. However, the distance between paired instruments in these studies may be more extensive than a research vessel's linear scale. The typical

period of imagery acquisition in SAILCOP is 20 s. In Table 1, we present a short description of the research missions resulting in the DASIO collection. In Appendix A, we also present the complete maps of the missions. In Table 2, we present the quantitative summary of the acquired collection on a per-mission basis and overall. Note that for the mission named AI-45, there is no TCC labeling information, whereas there were 113 observations made by an expert. In this mission, the observations of TCC were registered following the Russian standard of meteorological observations, which implies the estimation of cloudiness in tenths instead of oktas. Thus, these records are not standardized and are not included in the quantitative summary of the DASIO collection.

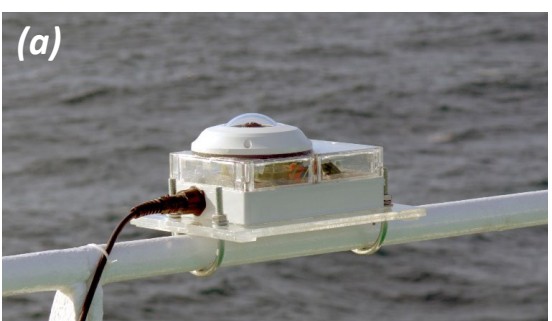 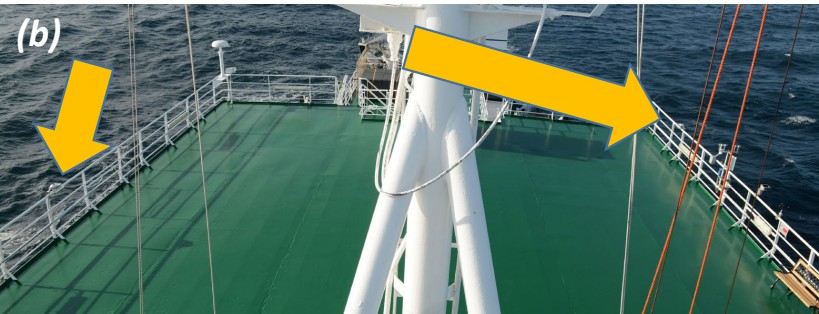

**Figure 2.** Optical package designed for the all-sky imagery acquisition: (**a**) Registering head, and (**b**) positions of registering heads mounted on the research vessel "Akademik Ioffe" in the research mission "AI-49" in June 2016.

**Table 1.** Scientific missions resulting in the Dataset of All-Sky Imagery over the Ocean (DASIO) collection of all-sky imagery over the ocean with the corresponding expert records of meteorological parameters.

| Mission Name | Departure | Destination | Route |
|---|---|---|---|
| AI-45 | 17 September 2014<br>Reykjavik, Iceland | 25 September 2014<br>Rotterdam, Netherlands | Northern Atlantic |
| AI-49 | 12 June 2015<br>Gdansk, Poland | 2 July 2015<br>Halifax, Canada | Northern Atlantic |
| ANS-31 | 16 December 2015<br>Colombo, Sri Lanka | 19 January 2016<br>Kaliningrad, Russia | Indian ocean, Red sea,<br>Mediterranean sea, Atlantic ocean |
| AI-52 | 30 September 2016<br>Gdansk, Poland | 3 November 2016<br>Ushuaia, Argentina | Atlantic ocean |
| ABP-42 | 21 January 2017<br>Singapore | 25 March 2018<br>Kaliningrad, Russia | Indian ocean, Red sea,<br>Mediterranean sea, Atlantic ocean |
| AMK-70 | 5 October 2017<br>Arkhangelsk, Russia | 13 October 2017<br>Kaliningrad, Russia | Northern Atlantic,<br>Arctic |
| AMK-71 | 24 June 2018<br>Kaliningrad, Russia | 13 August 2018<br>Arkhangelsk, Russia | Northern Atlantic<br>Arctic |
| AMK-79 | 13 October 2019<br>Kaliningrad, Russia | 5 January 2020<br>Montevideo, Uruguay | Atlantic ocean<br>Arctic |

**Table 2.** Quantitative summary of the DASIO collection in terms of the number of collected images and hourly meteorological observations.

| Mission Name | No. of Images | No. of Observations | No. of Observations per TCC (Total Cloud Cover) Value | | | | | | | | |
|---|---|---|---|---|---|---|---|---|---|---|---|
| | | | 0 | 1 | 2 | 3 | 4 | 5 | 6 | 7 | 8 |
| AI-45 | 8536 | 113 | - | - | - | - | - | - | - | - | - |
| AI-49 | 42,550 | 330 | 6 | 6 | 4 | 6 | 7 | 4 | 19 | 28 | 250 |
| ANS-31 | 79,748 | 349 | 28 | 29 | 21 | 19 | 40 | 21 | 20 | 30 | 141 |
| AI-52 | 218,890 | 374 | 38 | 22 | 13 | 25 | 24 | 20 | 22 | 36 | 174 |
| ABP-42 | 295,170 | 655 | 114 | 57 | 53 | 51 | 56 | 27 | 48 | 48 | 201 |
| AMK-70 | 30,336 | 68 | 0 | 1 | 2 | 1 | 1 | 4 | 11 | 16 | 32 |
| AMK-71 | 340,858 | 708 | 8 | 23 | 11 | 14 | 9 | 14 | 19 | 36 | 574 |
| AMK-79 | 139,194 | 355 | 15 | 23 | 17 | 28 | 26 | 16 | 35 | 50 | 145 |
| Total | 1,155,282 | 2839 | 209 | 161 | 121 | 144 | 163 | 106 | 174 | 244 | 1517 |

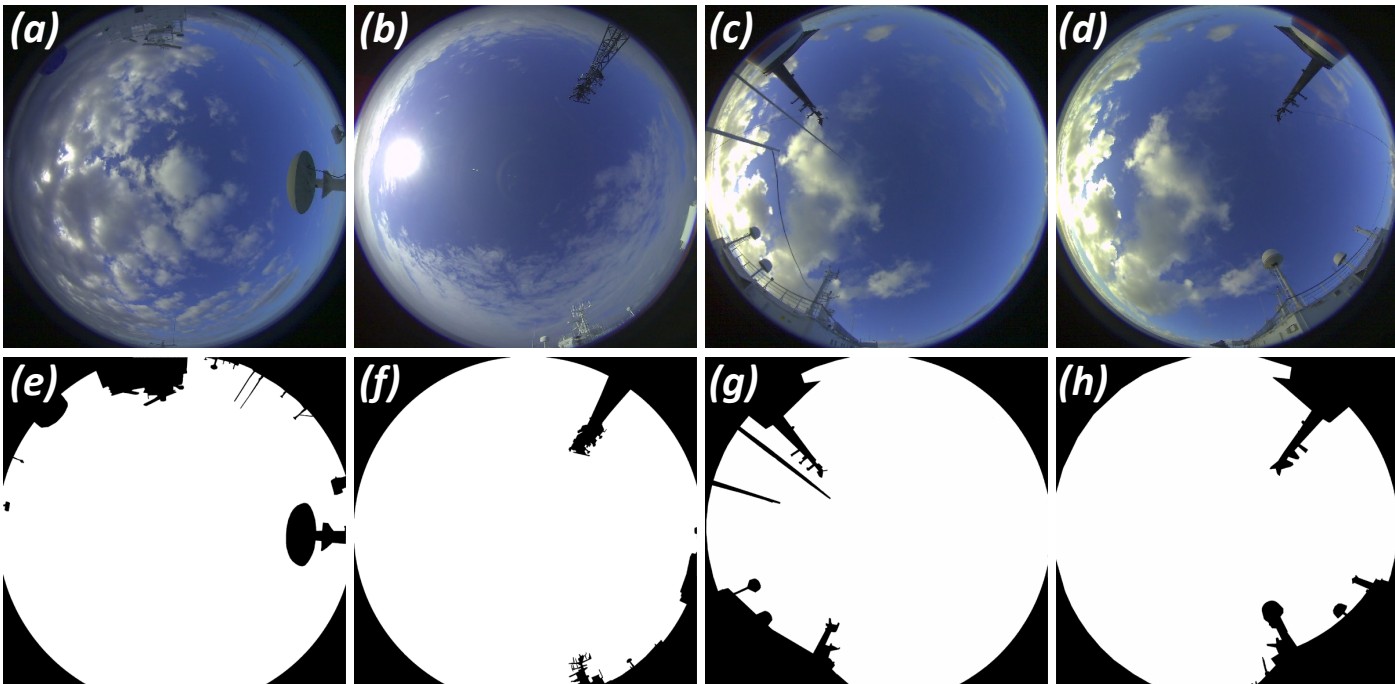

**Figure 3.** Examples of all-sky imagery (**a–d**) with the corresponding computation masks (**e–h**).

The DASIO collection obviously does not represent all the regions of the World Ocean. However, the regions of this dataset include the western, central, and eastern parts of the Indian Ocean as well as of northern Atlantic; the north-eastern, central, and south-western regions of the Atlantic ocean, Arctic ocean, and regions of the Mediterranean sea and the Red sea. Typical cloud types and cloud cover levels of these regions differ considerably, as mentioned in Section 1.2. Thus, using this dataset, one may assess the generalization ability of a data-driven scheme optimizing it on some subset of the DASIO collection and estimating it on a hold-out subset.

The DASIO collection will be released online in early 2021 with a corresponding announcement in the GitHub repository devoted to this study: https://github.com/MKr initskiy/TCCfromAllSkyImagery.

### 2.2. Data Preprocessing and Filtering

In this section, we present our approach for data preprocessing and filtering employed in this study. The DASIO collection includes some outliers, as shown in Figure 1. There are

also outliers caused by the cameras' malfunctioning, e.g., when the automatic algorithm for adjustment of white balance fails. This may happen in the early morning or in low-light conditions typically observed concurrently with stratocumulus clouds in high latitudes. In our study, we filter the collection by applying the Convolutional Variational Autoencoder model (CVAE) [67], see Section 3.1. This method allows us to filter out the snapshots that may be considered outliers. We inspect the results of the filtering to decide whether the images marked using our approach are indeed outliers. The criterion here is our expert understanding of whether the image represents its skydome scene the way that the major collection does. In Appendix A, we present a subsample of the images marked as outliers using the CVAE model. Some of the images are corrupted and may not be used as training examples. However, others may be considered valid. Expert inspection is still inevitable here.

For the application of the models of our study, we preprocess all the images of DASIO. First, we create a binary mask for each mount point of each mission. A mask is an image that is the same size as the DASIO examples. The goal of using masks is to mask the structures of a vessel. The masks for some of the missions are presented in Figure 3, along with the corresponding imagery. The whole set of masks for the imagery is included in DASIO. We also reduce the size of snapshots in order to fit our computational resources. The original size of DASIO examples is 1920 × 1920 px. In our study, we resize the images to 512 × 512 px. The masks are resized respectively.

For improving the generalization ability of the CVAE model and other data-driven models of our study, we apply the augmentation of data. Several studies have recently demonstrated the effectiveness of augmentation for increasing data-driven models' generalization ability [63,68–76]. In some studies, even the approach is presented for trainable data augmentation, which improves the training process [77–79]. In our study, we apply simple affine transformations along with weak elastic distortions described in [68] at the stage of image data augmentation.

All artificial neural networks of this study were trained on NVIDIA GPU using the PyTorch framework [80]. However, to improve the computational speed, we stopped using the torchvision implementations of imagery augmentations (a part of PyTorch project). We re-implemented all the transformations with pure PyTorch, so the augmentation is effectively performed on GPU, which is not always the case at the moment for native torchvision. As a result of this decision, we observed a speedup of the training by approximately four times. The code for the augmentations is integrated into our study's code, which is available on GitHub: https://github.com/MKrinitskiy/TCCfromAllSkyImagery. The distortions introduced at the augmentation time are stochastic by design: We sample the affine and elastic transformations' magnitudes. However, to preserve the consistency of the distorted imagery and the masks, we apply the same transformation to the masks as to the images.

Since we use artificial neural networks as the data-driven models in our study, we apply source data normalization recommended for the stabilization of training (see, e.g., [81]).

The sampling procedure is rather part of the experiment design thus, one may refer to Section 3 for the detailed description.

## 3. Machine Learning Models and Experiment Design

In this section, we present the design of the experiments in our study. We also present the competing architectures of artificial neural networks, which we employed for TCC retrieval from all-sky optical imagery over the ocean.

### 3.1. Filtering Outliers with Convolutional Variational Autoencoder (CVAE)

When filtering outliers we assumed that most of the typical examples of training dataset belong to some compact manifold in some feature space. Intuitively, one may expect a considerable difference between the cardinality of the set of all the possible RGB images of size 1920 × 1920 ($\sim 255^{10^7}$) and the amount of meaningful all-sky photographs. The latter

excludes meaningless chaotic images and real-world objects, persons, scenes of kinds other than all-sky, etc. Whether an example belongs to the manifold of a dataset or not may be determined by distance measuring, e.g., Euclidean distance in the original feature space described by R, G, and B components of all the pixels of an image. The dimensionality of this original feature space is $\sim 10^7$ in the case of DASIO examples. Thus, an effect of the high dimensionality of the examples takes place, which results in insignificant differences of distances between the examples that are close to each other ("seem similar" as images) or spaced apart ("seem dissimilar" as images). This effect is known as the "curse of dimensionality". One way to tackle this effect is a dimensionality reduction. The requirements for this mapping are simple: (i) It should preserve the relations between the examples of the training dataset (similar images should be projected to the points of a new feature space close to each other); (ii) at the same time, if the examples happened to be mapped close to each other in a new feature space, they should appear similar.

An artificial neural network is essentially a function that maps objects from one feature space (e.g., images) to another, the so-called feature space of hidden representations. There is a neural model capable of reducing the dimensionality of examples without losing much meaningful information of the examples, namely the Autoencoder (AE) [82–84]. Since the examples in our study are images, we exploit convolutional autoencoders. An autoencoder generally includes two functional parts: An encoder and a decoder. The encoder transforms the examples extracting meaningful features and mapping the examples into a hidden representation feature space. The dimensionality of this feature space is commonly lower than the original dimensionality of the examples. This is the case in our study. The decoder part decodes (reconstructs) the examples based on their hidden representations. The technical task for an autoencoder is to reproduce the examples with the lowest errors. Training an autoencoder is no different from training any other artificial neural network or other statistical models: One exploits a gradient-based optimization procedure for optimizing the loss function in the space of the neural network's parameters. In our study, we employ MSE as a reconstruction loss for the autoencoder model.

Though autoencoders are applied widely for anomaly detection, an autoencoder in its simple form (a.k.a. "vanilla" autoencoder in jargon) is not enough for filtering outliers. Only the second requirement is met in the case of a "vanilla" autoencoder: Examples that are close to each other in the hidden representation feature space are similar. However, the opposite is not true: Subsets of similar examples may be projected into clusters spaced apart. In our study, we overcome this issue by exploiting variational autoencoders [67] in the form of a CVAE. Variational autoencoders were shown to find a special kind of mapping that preserves the continuity of the feature space, meaning that the first requirement is met: Similar examples are projected to the close points of the hidden representation feature space. This property is achieved by introducing the assumption that each feature of the hidden representation of the dataset's examples is distributed normally. Technically this assumption can be met using an additional loss component of KL divergence between the sample distribution of hidden representation features and normal distribution. The loss component and a technical way for implementing this approach (known as "reparameterization trick") were proposed in [67]. The resulting loss function is presented in Equation (3) and includes reconstruction loss (MSE) along with the KL term. In Figure 4, we present the general architecture of the CVAE implemented in our study.

$$h_i = \mathcal{E}(x_i, \theta_e), \tag{1}$$
$$\hat{x}_i = \mathcal{D}(h_i, \theta_d), \tag{2}$$
$$\mathcal{L}(x_i, h_i, \hat{x}_i) = MSE(x_i, \hat{x}_i) + KL(h_i), \tag{3}$$

where $h_i$ is hidden representation of an example $x_i$; $\mathcal{E}(\cdot)$ and $\mathcal{D}(\cdot)$ are encoder and decoder parts of the autoencoder; $\theta_e$ and $\theta_d$ are parameters of the encoder and the decoder, respectively; and $\mathcal{L}(\cdot)$ is the CVAE loss function.

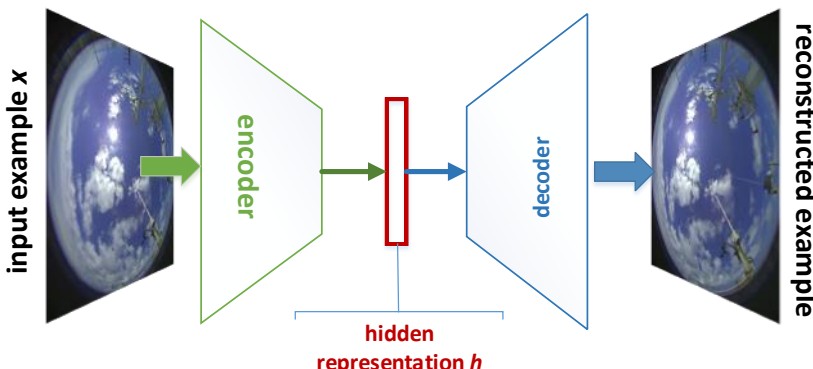

**Figure 4.** General architecture of the Convolutional Variational Autoencoder model (CVAE) model for filtering outliers. See more details in Appendix A.

In our study, we employ several approaches for improving the convergence of artificial neural networks. We employ the Adam optimizer [85] with the learning rate scheduling strategy named SGDR (Stochastic Gradient Descent with warm Restarts) presented in [86] that implies cosine annealing of learning rate with warm restarts. In addition to SGDR we apply exponential decay of the maximum learning rate. The resulting learning rate curve and typical training loss curve are presented in Figure A3 in Appendix A.

### 3.2. Neural Models for TCC Retrieval from All-Sky Imagery

Convolutional Neural Networks (CNNs) demonstrated a huge leap forward in image recognition [69,70,87], semantic segmentation [63,88–91], and other visual tasks. In most of the problems related to image processing, CNNs can achieve the highest quality with a gap unbridgeable by classic computer vision approaches. In some of the problems, CNNs reach human-like or even super-human quality today. Being data-driven models, CNNs seem to be perfect candidates for the application to TCC retrieval from all-sky imagery. In our study, we propose an advanced architecture of CNN motivated by the latest results in the field of neural architectures.

In contrast with a number of contemporary studies, we were looking for a state-of-the-art CNN architecture demonstrating the best results in a range of benchmark visual problems. In https://paperswithcode.com/, a perfect up-to-date collection of studies is presented. The resource is not of an academic sort, and it does not provide a comprehensive overview and does not perform benchmarking itself. However, it is still a tool for the selection of best-known approaches at the moment. Not to mention the set of benchmarks like MNIST [92], CIFAR-10/100 [70], Imagenet [93], and up to 165 others that are commonly considered by researchers when testing the proposed approaches and model architectures. The approaches presented in the state-of-the-art studies at the moment are of several types: Data augmentation strategies, strategies of training of neural networks, approaches for neural architecture search, meta-learning, and a set of approaches that pursue a goal of decreasing the computational overhead of neural networks without too much loss of the quality. One of the best neural designs we have come up with from studying the state-of-the-art architectural solutions, is the PyramidNet proposed in [94]. The key contribution of this work is the conscious design of the layers resulting in computational efficiency of the PyramidNet along with increased quality compared to other architectures with similar computational costs [95,96]. In our study, we tested both ResNet-like architecture and the one based on PyramidNet. No statistically significant difference was found. In this paper, we do not demonstrate the results of using ResNet-like architecture, as a thorough comparison of different architectures for TCC retrieval from all-sky imagery is not the topic of this study.

### 3.2.1. TCC Retrieval as Classification

As mentioned in Section 1.3, there are two formulations that we consider promising and valid for the TCC retrieval problem. The first and obvious problem design is classification, since the set of possible outcomes is finite and discreet. In this case, the target variable (TCC) is one-hot-encoded, that is, transformed into the form of a row-vector of $K$ elements, where $K$ is the number of classes of the problem. All elements of this vector are zero, and only one element of this vector equals 1, which corresponds to the label number. For example, the target vector for an image labeled as $TCC = 3$ okta is the following: $[0, 0, 0, 1, 0, 0, 0, 0, 0]$. As a result of this transformation, the one-hot-encoded target vector represents the probability distribution of a multi-variate target variable assuming that each component has Bernoulli distribution. In this formulation, a neural network approximates the parameters $p$ of Bernoulli distribution for each of the components of the target vector. In the Maximum Likelihood Estimation approach, this sequence of assumptions results in a loss function of multinomial cross-entropy (see Equation (6)). Note that a neural network that maps features of examples into the feature space of one-hot-encoded target variable is simply a parametric function $\mathcal{F}(x_i, \theta_{NN})$, where $\theta_{NN}$ are the parameters of the network. It is worth mentioning that the architecture of the neural network is not dependent on the loss function. Thus, one may consider the same architecture with a different formulation of the problem resulting in different loss functions and network behavior.

$$h_i = SoftMax(\mathcal{F}(x_i, \theta_{NN})), \tag{4}$$

$$SoftMax_k(z) = \frac{e^{z_k}}{\sum_{j=1}^{K} e^{z_j}}, \tag{5}$$

$$\mathcal{L}_{PC}(h_i, t_i) = -\sum_{j=1}^{K} t_{ij} * \log(h_{ij}), \tag{6}$$

where $x_i$ is the image raw data; $SoftMax(\cdot)$ is the function transforming a vector to a form meeting the requirements to represent a probability distribution (e.g., summation to one over the vector); $h_i$ is the estimates of parameters of Bernoulli distribution of one-hot-encoded target variable TCC denoted as $t_i$ (also known as "ground truth"); and $K$ is the number of classes ($K = 9$ for TCC retrieval following the recommendations of WMO). The subscript PC regards the formulation particularity, which is "Pure Classification" here. When a network is trained, the estimates $h_{ij}$ may be regarded as a measure of the probability of an event described by features $x_i$ to be of class $C_j$, where $\{C_j\}|_{j=1}^{K}$ is a set of classes of the problem.

The network described above is referred to hereafter as PNetPC, which stands for Pyramid Net for Pure Classification. The approach of classification using artificial neural networks is not new on the contrary, classification is one of the most common problems in machine learning. What is new in the presented approach is the formulation of the TCC retrieval as a classification.

### 3.2.2. TCC Retrieval as Ordinal Regression

As an alternative to classification, we consider the TCC retrieval as ordinal regression since TCC classes have a natural order. There are a few competing approaches for solving ordinal regression with artificial neural networks [97–103]. As mentioned above, the architecture of a network does not depend on its loss function. It defines the behavior of the network rather than its architecture. Thus, one may apply any of the approaches mentioned in the studies [64,97–103] with their own network architecture. In our study, we exploit PyramidNet within the approach of ordinal regression. Following the survey [64] and the original paper [98], we implemented our neural model to solve the problem of TCC retrieval within the approach of ordinal regression with no assumptions made on the distribution of target value. Alternatively, one may consider the formulation to be a soft classification: If an object $x_i$ is labeled as class $C_j$, it is inherently labeled as class $C_{j-1}$ and

so on down to $C_1$. Technically, this formulation implies only two minor changes: A new encoding of target variables and a new loss function. The new encoding implies that all the target vector elements of an example labeled as $C_j$ are equal to 1 in positions $j$ and lower. For example, the target vector for an image labeled as $TCC = 3$ okta are the following: $[1, 1, 1, 1, 0, 0, 0, 0, 0]$. As the problem is considered soft classification, each of the target vector elements is considered an independent random variable with Bernoulli distribution. Thus, the loss function, in this case, is just the sum of individual binary cross-entropy loss functions for all the $K$ components of the target vector, as shown in Equation (8).

$$h_i = \mathcal{F}(x_i, \theta_{NN}), \tag{7}$$

$$\mathcal{L}_{OR}(h_i, t_i) = -\sum_{j=1}^{K} \big(t_{ij} * \log(h_{ij}) + (1 - t_{ij}) * \log(1 - h_{ij})\big), \tag{8}$$

where the notation is the same as in Equation (6). The subscript OR regards the formulation particularity, which is "Ordinal Regression" here. Note that there is no $SoftMax(\cdot)$ in this case. However, each of the target vector's independent components still represents a Bernoulli parameter estimate. Thus, the activation function of the very last layer of the network $\mathcal{F}(x_i, \theta_{NN})$ needs to be sigmoid $\sigma(\xi) = \frac{1}{1+\exp(-\xi)}$ or a similar alternative (e.g., $tanh(\cdot)$ normalized accordingly).

The network described above is referred to hereafter as PNetOR, which stands for PyramidNet for Ordinal Regression. To the best of our knowledge, the problem of TCC retrieval has never been solved using any data-driven models within the approach of Ordinal Regression. As one may see in Section 4, ordinal regression delivers supreme quality compared to classification.

All the deep learning models described in this section are available in source code in the GitHub repository devoted to this study: https://github.com/MKrinitskiy/TCCfromAllSkyImagery.

*3.3. Experiment Design*

In this section, we propose the framework for assessing the quality and generalization ability of data-driven models in the problem of TCC retrieval. Given a model capable of estimating TCC for an expert-labeled all-sky image, one may compare the expert label with the model estimate. As mentioned in Section 1.3, $MSE$, $MAE$, correlation coefficient, or determination coefficient are questionable quality measures for data-driven models in the problem of TCC retrieval. We propose measuring the quality with accuracy (Equation (9)) in the case of a balanced test dataset.

$$Acc = \frac{\sum_{i=1}^{N} [T\hat{C}C_i = t_i]}{N}, \tag{9}$$

where $N$ is the number of test dataset examples; $T\hat{C}C_i$ is a model estimate of TCC; and $t_i$ is the expert-defined label for TCC (ground truth). In addition to accuracy, one may assess the quality using a common measure "less than or equal to one-okta error accuracy" ($Leq1A$, see Equation (10)). However, as mentioned in Section 1.3, this measure may be biased biased due to how the problem is designed.

$$Leq1A = \frac{\sum_{i=1}^{N} [|T\hat{C}C_i - t_i| \leq 1]}{N}. \tag{10}$$

Each configuration of data-driven models (PNetPC, PNetOR) is trained and evaluated multiple times (typically five to nine times due to high computational costs of CNNs) for estimating the uncertainty of the quality measures. Following the procedure for estimating the uncertainty of quality measures proposed in Maddox et al. in 2019 [104], we also saved a few snapshots (meaning the parameters set) of neural models trained in their state close to a quality plateau. This quality plateau is achieved using a constant learning rate while

training a model with the SGD optimizer, starting from a state named "pre-trained" [104], which is the final state of the model assessed in this study. We then use quality estimates of these snapshots as additional realizations of the random variables, namely quality measures. We assess all the quality measures' uncertainty as a confidence interval of 95% confidence level, assuming that the measures are distributed normally.

In this section, we employ the same approaches as in Section 3.1 for improving the convergence of artificial neural networks. We use the Adam optimizer [85] with a SGDR learning rate scheduling strategy [86] and exponential decay of the maximum learning rate. The resulting learning rate curve and typical learning curves for PNetOR are presented in Figure A4 in Appendix A. We do not show the learning curves for PNetPC in this paper. They are similar to the ones of PNetOR with the only difference being that PNetOR delivers slightly better quality, as one can see in the Results Section 4.

### 3.4. Datasets Balancing and Subsampling Procedures

The DASIO collection is strongly unbalanced: The number of examples labeled with 8 okta is approximately 50% (see the region-agnostic TCC distribution in Figure 5). In our study, all the subsets (training and test) are balanced the following way: We randomly subsample the subset of the examples labeled as 8 okta. The number of 8-okta subsample examples is the average number of the rest classes (marked with orange lines in Figure 5). The other classes are subsampled or oversampled to reach the same number of examples. This target-balancing procedure is applied in all the experiments described below.

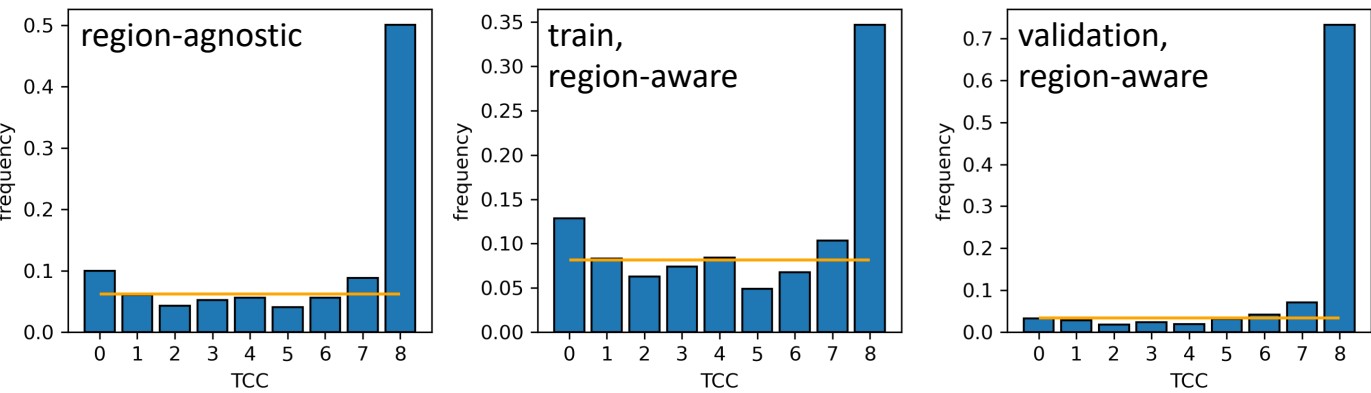

**Figure 5.** Histograms of the empirical distributions of TCC in different scenarios: Region-agnostic (train and test distributions are the same) and region-aware. In all of the three cases, orange lines denote the average levels for 0–7 okta classes used for target-balancing each of the subsets; 8-okta class is downsampled to this level following the procedure proposed in Section 3.3. See details on exact numbers of cases in Appendix A, Table A2.

#### 3.4.1. Region-Agnostic Scenario

To compare different data-driven algorithms for TCC retrieval, we propose training them on the same training subset and assessing their quality on the same test subset of imagery. In our study, we applied this procedure for comparing the performance of the two proposed models, namely PNetPC and PNetOR. In this scenario, we do not impose any restrictions on the regions where the data were collected for train and test subsets. We name this scenario "region-agnostic" hereafter. Note, however, that objects of the DASIO collection may be correlated when being close in time. In time series analysis, there is a known effect of naturally correlated examples that are temporally related. This is especially the case when the imagery is taken with the period of order of 20 s. This is considerably shorter than the typical period for decorrelating of spatial cloud characteristics, which was estimated to be ~15 min [65]. Thus, the common data science procedure of completely random train-test split produces strongly correlated (i.e., not independent) subsets, and thus the estimates of the quality are ultimately too optimistic. This effect strongly influenced

the results presented in [57]. Instead of entirely random subsampling, one needs to employ the procedure of block-split. In our study, we consider each day of observations as a block of data. This way, we overcome the issue of an auto-correlated time series of TCC and all-sky imagery. Currently, the DASIO collection contains 250 days of observations. Thus, there are 250 daily blocks. In the case of non-restricted splitting procedure, we split these 250 blocks into train and test subsets. As the classes' fractions may vary depending on the split realization, we did our best to select the split that produces the most equally distributed train and test subsets. Particularly, we sample the blocks for subsets with a subsequent computation of KL divergence between the TCC distributions produced. After 10,000 sampling attempts, we consider the best split that results in the lowest KL divergence. This way, the fractions of TCC classes are almost the same in train and test subsets. Although we still cannot guarantee the absence of covariate shift this way, at least we guarantee a low shift of a target's distribution.

### 3.4.2. Region-Aware Scenario

For assessing the generalization ability of data-driven models, we propose comparing the quality estimates evaluated within the region-agnostic approach described above with the ones achieved with a different strategy of train-test split. The region-aware approach implies the restrictions on the regions of imagery acquisition. In this scenario, a strong covariate shift is expected, as mentioned in Section 1.2 on the climatology of clouds. In particular, we train the models on the subset limited by latitudes from 45°S to 45°N. In this scenario, we assess the quality using the data subset acquired to the north of 45°N. As one may see in the maps of the missions which contributed to the DASIO collection, the regions selected for the training set this way include the central Atlantic, the Mediterranean and Red seas, and all the observations in the Indian ocean. The regions selected for the test set include the northern Atlantic (the section along the 60°N and coastal regions) and the Atlantic sector of the Arctic ocean. In Figure 5, the distributions of TCC values are presented for the region-agnostic and region-aware scenarios.

### 3.5. Quantitative Characteristics of Datasets

Data-driven models, particularly deep learning ones, are known to be strongly dependent on the amount of data. The size and diversity of a training set are crucial for both the generalization ability and expressive power of a data-driven model. A validation set's size and diversity are crucial for the reliability of the model's quality estimates. The number of observations of the whole DASIO collection is 2'839, and almost half of this amount are observations with $TCC = 8$ (see Table 2). This low amount of data points may be considered highly insufficient for deep learning models described in Sections 3.2.1 and 3.2.2. For increasing the number of data points for both training and validation, we assumed that images close (temporally) to expert observations have the same TCC value. This approach is valid due to the consideration of strongly correlated examples close to each other in terms of acquisition time. The typical period for decorrelating spatial cloud characteristics is considered to be ~15 min [65]. In our study, we considered all the images acquired within ±10 min of the time of expert observation as having the same TCC as registered by the expert. This procedure extends the subsets that were used for training and testing in both region-agnostic and region-aware scenarios. The quantitative characteristics of the datasets are presented in Table 3. Note that the numbers of data points in the "balanced train" and "balanced test" columns are less than the number of examples of the whole training and test sets due to the balancing procedure described earlier in Section 3.4.

**Table 3.** Number of examples in training and test subsets in both region-agnostic and region-aware scenarios.

| Scenario | Train | Test | Balanced Train | Balanced Test |
|----------|-------|------|----------------|---------------|
| Region-agnostic | 236,131 | 78,387 | 130,302 | 40,410 |
| Region-aware | 170,665 | 132,269 | 125,055 | 39,321 |

## 4. Results and Discussion

In this section, we present the results of the numerical experiments described in Section 3. For each of the data-driven models proposed in this study, we performed the training and quality assessment several times (typically five to nine times) to estimate the uncertainty of the quality measures. The results in this section are presented in the following manner: First, in Table 4, we present the results of the PyramidNet-based models PNetPC and PNetOR in the region-agnostic scenario for assessing the influence of the problem formulation. Then, in Table 5, we present the results of the PNetOR model in both the region-agnostic and region-aware scenarios to assess the generalization ability in the case of a strong covariate shift.

In our study, we also performed the optimization and quality assessment of some of the known schemes mentioned in Section 1.1 of the Introduction for a comparison with ours. This analysis was performed within the same framework as PNetPC and PNetOR. As mentioned in Section 1.1, each of the schemes is essentially data-driven and has its own parameters (at least one) subjected to optimization based on a training subset. The schemes used for comparison are the ones that may be considered a baseline. In particular, we assessed the following algorithms:

- The algorithm proposed by Long et al. in [38,39], which we name "RBR" hereafter after the main index proposed in these studies (red-to-blue ratio);
- The algorithm proposed by Yamashita et al. in [42,43,51], which we name "SkyIndex" hereafter after the index proposed in these studies.

As with the models proposed in this study (see Section 3.3), we assessed the uncertainty of the schemes RBR and SkyIndex by training them and estimating their quality multiple times. The schemes RBR and SkyIndex are not as highly demanding computationally as PNetPC and PNetOR. Thus, we were able to repeat the procedure 31 times. The results of this quality assessment are presented in the Table 5.

**Table 4.** Quality estimates for PNetPC (Pyramid Net for Pure Classification) and PNetOR (Pyramid-Net for Ordinal Regression) in the region-agnostic scenario.

| Model | $Acc_{train}$ | $Acc_{val}$ | $Leq1A_{train}$ | $Leq1A_{val}$ |
|-------|---------------|-------------|-----------------|----------------|
| PNetPC | $46.52 \pm 4.6\%$ | $41.43 \pm 2.3\%$ | $86.52 \pm 1.87\%$ | $85.7 \pm 2.1\%$ |
| PNetOR | $46.44 \pm 0.38\%$ | $42.38 \pm 0.97\%$ | $88.4 \pm 0.23\%$ | $84 \pm 0.18\%$ |

From the results presented in Table 4, we can see that PNetOR is slightly superior. However, we need to mention that this superiority is not statistically significant, at least for such a small number of runs. In addition, in terms of $Leq1A$, PNetPC is slightly better than PNetOR. It is worth mentioning that state-of-the-art expert-designed schemes for TCC retrieval demonstrate a considerably lower quality level: In no fair conditions, $Acc$ for them exceeds 30% [49,57]. In the scenarios employed in this study, the estimated accuracy did not exceed 27%. In Table 5, we present more detailed quality characteristics of the RBR and SkyIndex's schemes estimated within the proposed framework.

**Table 5.** Quality estimates for PNetOR, red-to-blue ratio (RBR) [39], and SkyIndex [42] in different scenarios.

| Scheme (Model) | Scenario | $Acc_{Train}$ (Percent) | $Acc_{val}$ (Percent) | $Leq1A_{Train}$ (Percent) | $Leq1A_{val}$ (Percent) | $RMSE_{Train}$ (okta) | $RMSE_{val}$ (okta) |
|---|---|---|---|---|---|---|---|
| RBR | region-agnostic | $25.7 \pm 0.2$ | $25.9 \pm 0.5$ | $57.5 \pm 0.2$ | $61.4 \pm 0.4$ | $2.18 \pm 0.01$ | $2.18 \pm 0.02$ |
| RBR | region-aware | $24.7 \pm 1.0$ | $26.3 \pm 0.2$ | $59.6 \pm 0.2$ | $54.8 \pm 0.4$ | $2.12 \pm 0.01$ | $2.37 \pm 0.01$ |
| SkyIndex | region-agnostic | $25.8 \pm 0.3$ | $25.9 \pm 0.4$ | $57.5 \pm 0.3$ | $61.3 \pm 0.4$ | $2.18 \pm 0.01$ | $2.18 \pm 0.02$ |
| SkyIndex | region-aware | $24.7 \pm 0.2$ | $26.4 \pm 0.3$ | $59.6 \pm 0.3$ | $54.7 \pm 0.5$ | $2.12 \pm 0.01$ | $2.37 \pm 0.02$ |
| PNetOR | region-agnostic | $46.4 \pm 0.4$ | $42.4 \pm 1.0$ | $88.4 \pm 0.2$ | $84.0 \pm 0.2$ | $1.01 \pm 0.02$ | $1.24 \pm 0.02$ |
| PNetOR | region-aware | $48.4 \pm 0.7$ | $36.6 \pm 0.5$ | $90.0 \pm 0.4$ | $77.4 \pm 0.6$ | $0.95 \pm 0.04$ | $1.62 \pm 0.06$ |

According to the considerations set out in Section 1.3, RMSE may be a biased quality metric in the case of an ordered finite set of outcome values (e.g., TCC). In Table 5, these metrics are provided for a comparative purpose only.

From the results presented in Table 5, one can see a considerable difference in the gaps between training accuracy and its test estimate in different sampling scenarios. In the region-agnostic scenario, the gap is $\sim 4\%$, while in the region-aware scenario, the gap is $\sim 12\%$. We need to remind the reader that this gap is inevitable, and only a perfect model would deliver the same quality on the test set as on the training set (sometimes even better in some particular cases). However, as mentioned in Section 3.3 (Experiment design), we assess a data-driven model's capability to generalize in terms of these gaps. This means if the gap increases significantly, then the typical covariate shift between regions is too strong, and the combination of model flexibility and data variability guides a researcher to collect more data for the model to be applicable in a new region with an acceptable level of confidence. In the presented cases, a significant difference is noticeable.

One possible reason for that may be a substantial disproportion of TCC classes in the region-aware test subset. One may conduct a hypothetical experiment of assessing the quality of a scheme that predicts 8 oktas disregarding the real skydome scene characteristics. The accuracy score of such an algorithm in high latitudes would be outstanding. However, this quality measure would be definitely unreliable. It is worth mentioning that our CNN-based approach is still superior to the previously published results even in this worst-case scenario of a strongly unbalanced test subset.

One may propose another possible reason for the significant increase in the quality gap between the region-agnostic and region-aware scenarios. In a routine study involving machine learning, the common reason for such a considerable quality gap would be overfitting. Indeed, convolutional neural networks may be engineered to have such high expressive power that they would be able to "memorize" training examples, demonstrating low quality on the test set. However, the concept of overfitting relies on the relations between the quality estimates on training and test samples that are drawn from the same distribution or at least on the samples demonstrating weak signs of covariate shift. This is the case in the region-agnostic scenario, where the sampling and subsetting procedures described in Section 3.4.1 ensure that the data of training and test sets was at least acquired in similar conditions. In the region-aware scenario, training and test sets should be treated as having a substantial covariate shift, as discussed in Section 1.2. Thus, the concept of overfitting per se cannot be applied in this case. On the contrary, such a significant increase in the quality gap between the region-agnostic and region-aware scenarios suggests a strong covariate shift in the latter case.

One may also notice that the thresholding-based schemes like RBR or SkyIndex demonstrate an outstanding generalization ability: The quality drops from training to test subsets do not differ much between region-agnostic and region-aware scenarios. However, the expressive power of these models is weak, and even in the worst-case scenario the PNetOR model demonstrates significantly higher quality in terms of all the presented measures.

## 5. Conclusions and Outlook for Future Study

In this study, we presented the DASIO collected in various regions of the world ocean. Some of the imagery is accompanied by cloud characteristics observed in situ by experts. We demonstrated that there was a strong covariate shift in this data due to the natural climatological features of the regions represented in DASIO. We proposed a framework for the systematic study of automatic schemes for total cloud cover retrieval from all-sky imagery. We also proposed quality measures for the assessment and comparison of the algorithms within this framework. We also presented a novel data-driven model based on convolutional neural networks of state-of-the-art architecture capable of performing the task of TCC retrieval from all-sky optical imagery. We demonstrated alternatives for the formulation of the TCC estimation problem: Classification and ordinal regression. The results demonstrated a slight superiority of the version based on ordinal regression. We assessed the quality measures of the proposed models guided by the framework we introduced. These data-driven models based on convolutional neural networks demonstrated a considerable improvement in TCC estimation quality compared to the schemes known from the previous studies.

Most importantly, in this study we proposed an approach for testing the generalization ability of data-driven models in TCC retrieval from the imagery of the DASIO collection. We demonstrated a considerable drop in the generalization ability expressed in terms of the gap between the train and test quality measures in the scenario implying close distributions of objects' features compared to the scenario characterized by a strong covariate shift ($\sim$4 and $\sim$12 percentage points accordingly). The considerable quality gap in the former scenario (namely "region-aware") may indicate that specialized algorithms with high expressive power applied in regionally limited conditions may practically be of higher demand than the ones characterized by high generalization ability along with low accuracy. There is also an indication of a strong need for additional field observations in a greater variety of regions of the world ocean with the concurrent acquisition of all-sky imagery. This would help enrich the part of the DASIO collection with low TCC values.

In addition, although convolutional neural networks demonstrate impressive results in the problems of low-noise images recognition with strongly curated labeling information, in the TCC retrieval problem, a CNN model based on state-of-the-art architecture does not deliver that high quality being trained with the most-used tricks aimed for the improvement of generalization ability and training stabilization. The quality drop may also be due to noisy source data or uncertain labels. The issue of the labels' uncertainty will be addressed in our future studies.

Several unanswered questions remain that may be addressed regarding data-driven models in the problems of retrieving cloud characteristics from the DAISO data. We will tackle the problems of estimating the cloud base height since the SAILCOP optical package allows us to acquire paired imagery. We also will assess the uncertainty of expert labels for TCC and other characteristics. This may be possible due to the presented CVAE model's capability to preserve the proximity relations between examples of the dataset. One may assess the uncertainty of expert labels for similar examples identified with the use of CVAE. We will also continue the neural architecture search to find a CNN-based model that is balanced in quality and generalization. We will reproduce the results of all the previously published schemes for automatic TCC retrieval and assess their ability to generalize within the framework proposed in this study. We will also formulate a similar approach for the assessment of automated algorithms for cloud type identification.

**Author Contributions:** Conceptualization, project administration, M.K. and S.G.; methodology, software, validation, formal analysis, data curation, visualization, M.K.; investigation, M.K., M.A., P.V., A.S., A.G., N.K.; funding acquisition, resources and supervision, S.G.; writing—original draft preparation, M.K. and M.A.; writing—review and editing, M.K. and N.K. All authors have read and agreed to the published version of the manuscript.

**Funding:** This work was undertaken with financial support by the Russian Ministry of Science and Higher Education (agreement №05.616.21.0112, project ID RFMEFI61619X0112).

**Institutional Review Board Statement:** Not applicable.

**Informed Consent Statement:** Not applicable.

**Data Availability Statement:** The data presented in this study will be available in early 2021 here: https://dasio.ocean.ru/. The release will be announced in the GitHub repository devoted to this study: https://github.com/MKrinitskiy/TCCfromAllSkyImagery.

**Acknowledgments:** We are deeply indebted to Svyatoslav Elizarov for his contribution as a consultant on deep learning techniques.

**Conflicts of Interest:** The authors declare no conflict of interest.

## Abbreviations

The following abbreviations are used in this manuscript:

| | |
|---|---|
| ENSO | El Niño Southern Oscillation |
| TCC | Total Cloud Cover |
| NAO | North Atlantic Oscillation |
| PDO | Pacific Decadal Oscillation |
| ITCZ | Intertropical Convergence Zone |
| VOS | Voluntary Observing Ships |
| ICOADS | International Comprehensive Ocean-Atmosphere Data Set |
| WMO | World Meteorological Organization |
| DASIO | Dataset of All-Sky Imagery Over the Ocean |
| CVAE | Convolutional Variational Autoencoder |
| (R)MSE | (root) mean squared error |
| KL | Kullback–Leibler (divergence) |
| SGDR | stochastic gradient descent with warm restarts [86] |
| PNetPC | PyramidNet for Pure Classification |
| PNetOR | PyramidNet for Ordinal Regression |
| RBR | red-to-blue ratio |

## Appendix A

In this section, we present the materials that we consider important for the reproducibility of our study, though not essential for the main sections of the paper. In Figure A1, we present the map of all of the research cruises conducted so far that contributed to the DASIO collection. In Figure A2, we present examples of the DASIO collection marked as outliers by the CVAE model used for data filtering presented in Section 3.1.

In Table A1, we present the details of the CVAE model's architecture for filtering DASIO outliers. The model is built using ResNet [96] building blocks (namely, `residual_block` and `identity_block`) to improve training stability and increase the capability for the training itself. More details of the CVAE model's implementation are available in the source code of our study at GitHub: https://github.com/MKrinitskiy/TCCfromAllSkyImagery.

In Figure A3, we also present an example of learning curves for one of the training runs of the CVAE model.

In Figure A4, we present the details of PNetOR model training in different data split scenarios characterized by different covariate shift strengths. We do not show the learning curves of all the runs that were performed in order to assess the uncertainty of the quality estimates. Instead, we demonstrate only one typical run per scenario.

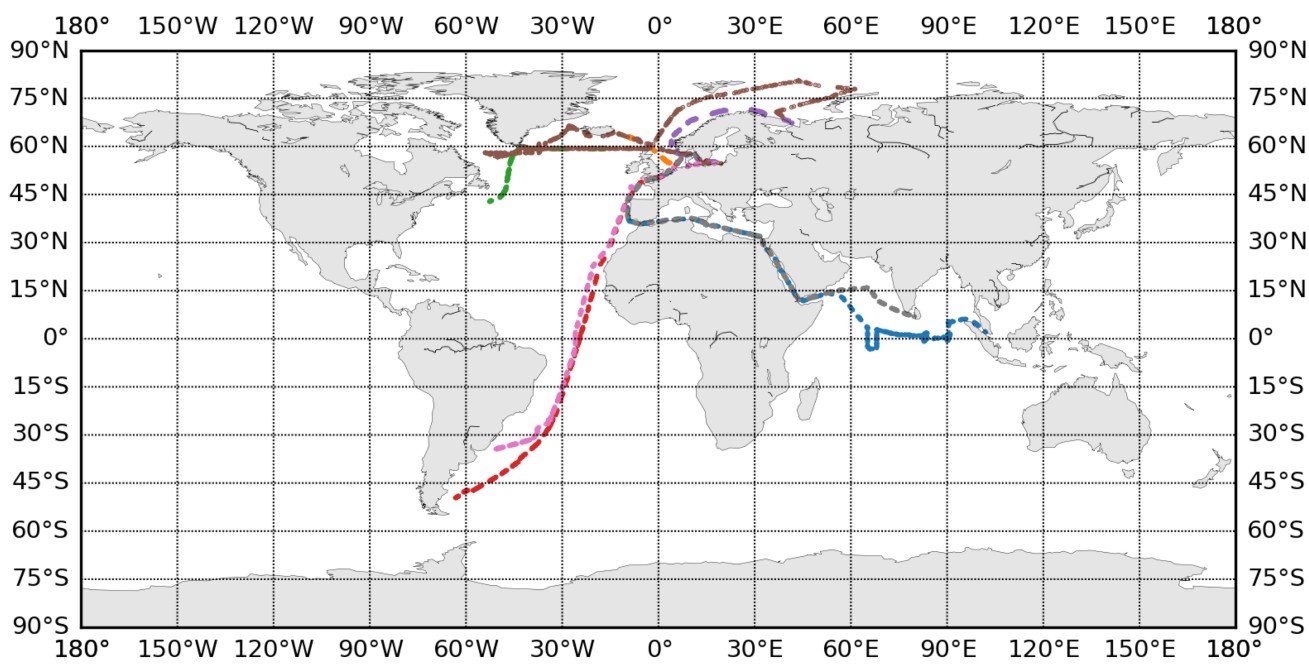

**Figure A1.** Routes of scientific missions resulting in DASIO collection.

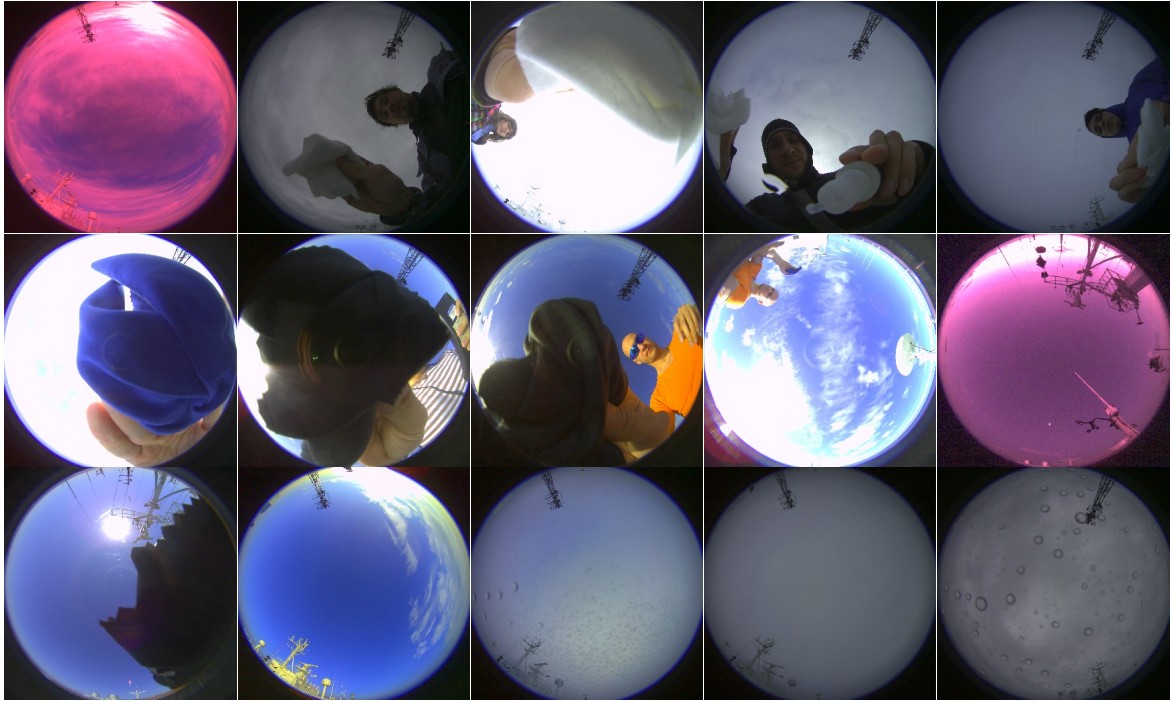

**Figure A2.** Examples from the DASIO collection marked as outliers by the CVAE model used in this study.

**Table A1.** CVAE architecture details.

| Block Name | Block Type | Inputs | Input Size | Output Size |
|---|---|---|---|---|
| encoder_input | input | - | 256,256,3 | 256,256,3 |
| mask_input | input | - | 256,256,3 | 256,256,3 |
| residual_block_1c | residual_block | encoder_input | 256,256,3 | 256,256,32 |
| identity_block_1i | identity_block | residual_block_1c | 256,256,32 | 256,256,32 |
| residual_block_2c | residual_block | identity_block_1i | 256,256,32 | 128,128,64 |
| identity_block_2i | identity_block | residual_block_2c | 128,128,64 | 128,128,64 |
| residual_block_3c | residual_block | identity_block_2i | 128,128,64 | 64,64,128 |
| identity_block_3i | identity_block | residual_block_3c | 64,64,128 | 64,64,128 |
| residual_block_4c | residual_block | identity_block_3i | 64,64,128 | 64,64,128 |
| identity_block_4i | identity_block | residual_block_4c | 64,64,128 | 32,32,256 |
| residual_block_5c | residual_block | identity_block_4i | 32,32,256 | 32,32,256 |
| identity_block_5i | identity_block | residual_block_5c | 32,32,256 | 16,16,512 |
| identity_block_6i | residual_block | residual_block_5c | 32,32,256 | 16,16,512 |
| residual_block_6c | identity_block | identity_block_6i | 16,16,512 | 8,8,1024 |
| identity_block_7i | residual_block | residual_block_6c | 8,8,1024 | 8,8,1024 |
| residual_block_7c | identity_block | identity_block_7i | 8,8,1024 | 4,4,1024 |
| enc_gap2d | GlobalAveragePooling2D | identity_block_7i | 8,8,1024 | 1024 |
| bottleneck | fully-connected | enc_gap2d | 1024 | 512 |
| z_mean_fc | fully-connected | bottleneck | 512 | 512 |
| z_var_fc | fully-connected | bottleneck | 512 | 512 |
| z_sampling | normal sampling | z_mean_fc,z_var_fc | (512), (512) | 512 |
| dec_input | fully-connected | z_sampling | 512 | 1024 |
| dec_reshape | Reshape | dec_input | 1024 | 32,32,1 |
| dec_identity_block_1i | identity_block | dec_reshape | 32,32,1 | 32,32,1024 |
| dec_identity_block_2i | identity_block | dec_identity_block_1i | 32,32,1024 | 32,32,1024 |
| dec_upsampling_1u | UpSampling2D | dec_identity_block_2i | 32,32,1024 | 64,64,1024 |
| dec_identity_block_3i | identity_block | dec_upsampling_1u | 64,64,1024 | 64,64,512 |
| dec_identity_block_4i | identity_block | dec_identity_block_3i | 64,64,512 | 64,64,512 |
| dec_upsampling_2u | UpSampling2D | dec_identity_block_4i | 64,64,512 | 128,128,512 |
| dec_identity_block_5i | identity_block | dec_upsampling_2u | 128,128,512 | 128,128,256 |
| dec_identity_block_6i | identity_block | dec_identity_block_5i | 128,128,256 | 128,128,256 |
| dec_upsampling_3u | UpSampling2D | dec_identity_block_6i | 128,128,256 | 256,256,256 |
| dec_identity_block_7i | identity_block | dec_upsampling_3u | 256,256,256 | 256,256,256 |
| dec_identity_block_8i | identity_block | dec_identity_block_7i | 256,256,256 | 256,256,256 |
| dec_identity_block_9i | identity_block | dec_identity_block_8i | 256,256,256 | 256,256,128 |
| dec_identity_block_10i | identity_block | dec_identity_block_9i | 256,256,128 | 256,256,128 |
| dec_conv2d_out | Conv2D | dec_identity_block_10i | 256,256,128 | 256,256,3 |
| dec_pw_norm | min-max normalization | dec_conv2d_out | 256,256,3 | 256,256,3 |
| masking | element-wise multiplication | mask_input, dec_pw_norm | (256,256,3), (256,256,3) | 256,256,3 |

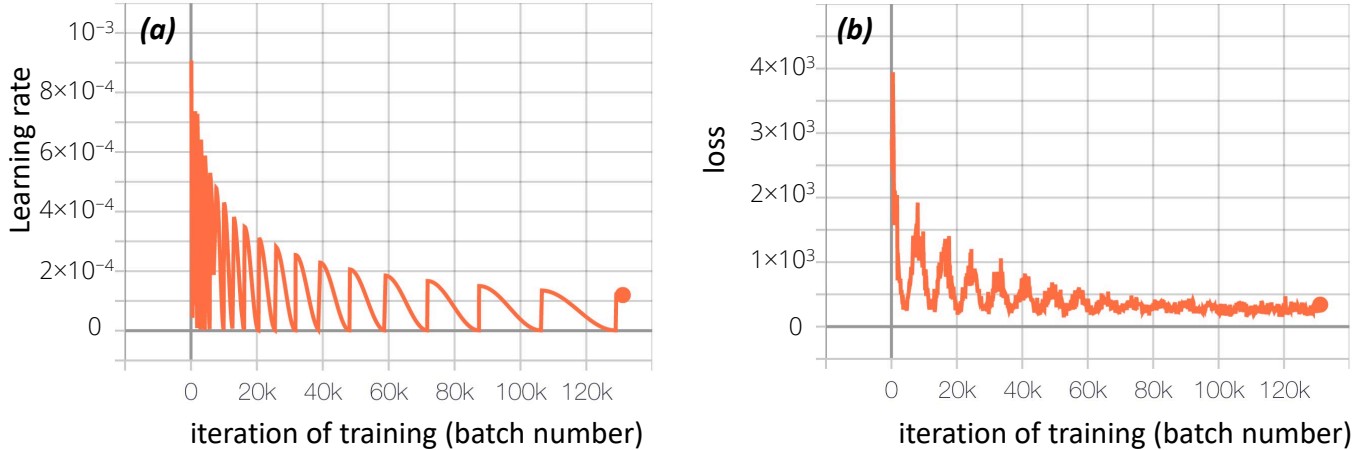

**Figure A3.** Training details of the CVAE model: (**a**) The learning rate scheduling strategy implies SGDR (Stochastic Gradient Descent with warm Restarts) [86] with exponential decay of the maximum value of the learning rate and (**b**) the typical learning curve (estimated on hold-out test subset of DASIO).

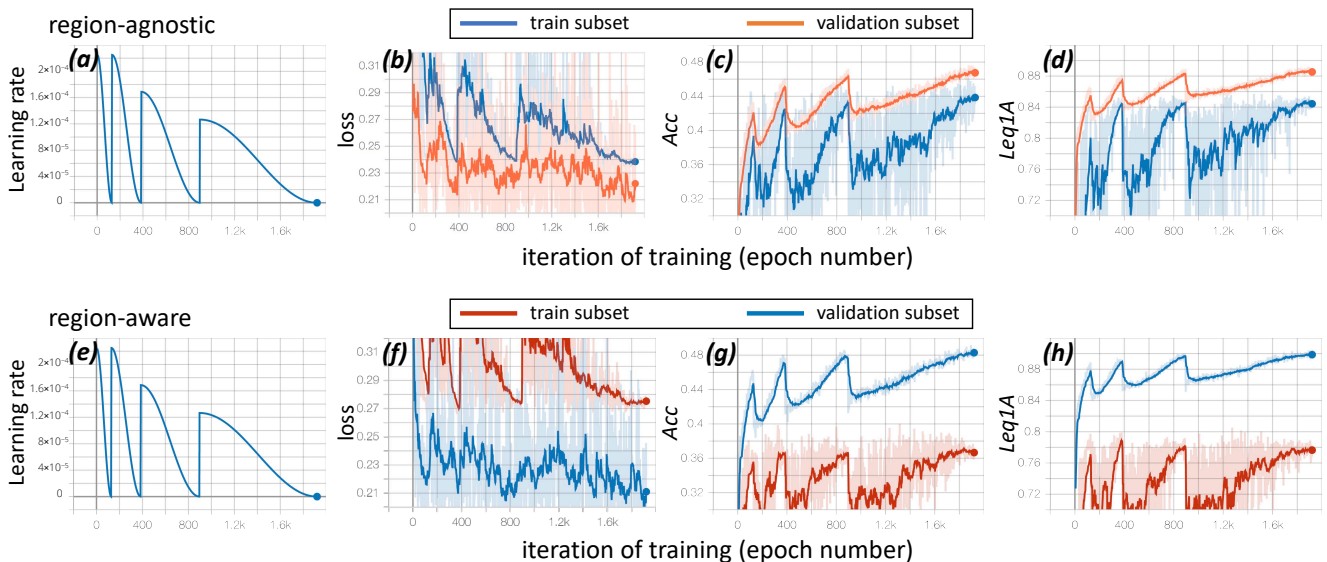

**Figure A4.** Training details of the data-driven models of our study: (**a**–**d**) Learning curves for the region-agnostic scenario and (**e**–**h**) learning curves for the region-aware scenario.

**Table A2.** Numerical values of the histograms presented in Figure 5 of the empirical distributions of TCC in different scenarios: Region-agnostic and region-aware. Here *TBLvl* (target-balancing level) is the average level for 0–7 okta classes that is used for target-balancing each of the subsets; the 8-okta class is downsampled to this level following the procedure proposed in Section 3.3.

| Scenario, Subset | *TBLvl* | No. of Observations per TCC Value | | | | | | | | |
|---|---|---|---|---|---|---|---|---|---|---|
| | | 0 | 1 | 2 | 3 | 4 | 5 | 6 | 7 | 8 |
| region-agnostic train | 14,478 | 20,834 | 14,573 | 10,888 | 12,289 | 13,078 | 9028 | 14,363 | 20,774 | 120,304 |
| region-agnostic test | 4490 | 5898 | 4508 | 2722 | 4080 | 4754 | 2986 | 4437 | 6539 | 42,463 |
| region-aware train | 13,895 | 20,380 | 14,449 | 11,132 | 12,595 | 14,096 | 8526 | 12,817 | 17,167 | 59,503 |
| region-aware test | 4369 | 1876 | 4152 | 2478 | 3774 | 3736 | 3488 | 5983 | 9468 | 97,314 |

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
