# Peer review of "On the Generalization Ability of Data-Driven Models in the Problem of Total Cloud Cover Retrieval"

_remotesensing, doi:10.3390/rs13020326_

Round 1
Reviewer 1 Report
A clear indication of how many images are contained in the DASIO collection, how many and what types of labels are made by the experts is required. The length of the dataset used and the size of the training dataset are not fully understood. Conditions for starting and ending optical observation of clouds, such as the sun's elevation threshold, must be specified.
The annotation states that the model provides the best results for estimating the cloud cover, but there is no comparison in the text. A convincing argument would be to test the model on the same set of images used in similar studies. It is also useful to present data from Table 2 as RMSE estimates of the total cloud cover and their comparison with the climatic variability of cloud cover.
1) page 2, line 64: no number of reference
2) page 4, line 134: to the list of noise sources, the authors should add a swinging camera installed on board the vessel and the need to correct recorded images. An adjustment to the height of the sun must also be taken into account
3) page 4, line 144: the purpose to use of the three (train, validation and test) datasets is not clear
4) page 5, line 226: no number of reference
5) page 7, line 306: the dependence of the accuracy of the models on the geographic region indicates their low generalisation capacity, possibly due to under- or over-training
6) page 8, line 364: it is not clear why the 512x512 image size is optimal in terms of performance improvement
7) page 9, line 381: typo "transformations transformations"
8) table 1, line 2: typo "12.16.2015"
9) table 2: the type of characteristic after the plus/minus symbol is not clear
10) page 14, line 537: typo "see eq. ??"
Reviewer 2 Report
The article is devoted to the current topic related to tcc recovering and written in a competent language. Interesting conclusions or many scientific fields are obtained, results can be very interesting for observational astronomy (observational time planning). The manuscript must be published after correction of following remarks:
Abstract
2 line. What is human-designed algoritms? I suggest to remove «human-designed».
Introduction
We can recommended to authors indicate in the article the practical significance of the definition of tcc, for example, for optical astronomy. Tcc determines the observation capability and the number of hours of observations. In this direction, engaging deep learning techniques is vital. Deep learning techniques for tcc recovering and prediction is especially important on short time scales. Perhaps the authors will consider it correct to use references:
- Baran, Á., Lerch, S., El Ayari, M. et al. Machine learning for total cloud cover prediction. Neural Comput & Applic (2020). https://doi.org/10.1007/s00521-020-05139-4
- … Kovadlo P.G. et. al. Atmospheric parameters at the 6-m Big Telescope Alt-Azimuthal site / MNRAS. V. 493. I.1. 2020. P.723 — 729. (http://academic.oup.com/mnras/article-abstract/493/1/723/5740325?redirectedFrom=fulltext)
- Ye Q.-Z. and Chen S.-S. The ultimate meteorological question from observational astronomers: how good is the cloud cover forecast? / MNRAS. 428. P. 3288 — 3294. 2013.
-Joschua A Hellemeier, Rui Yang, Marc Sarazin, Paul Hickson Weather at selected astronomical sites – an overview of five atmospheric parameters / Monthly Notices of the Royal Astronomical Society, Volume 482, Issue 4, February 2019, Pages 4941–4950, https://doi.org/10.1093/mnras/sty2982
Line 39. Although applying reanalysis data underestimate tcc values, it is shown that low-frequency temporal changes in tcc (reanalysis) are consistent with tcc estimated from observtions on the meteorological station (http://ru.iszf.irk.ru/images/1/1f/JSTP_6_1_2020_102-107.pdf)
Line 537. There is error in «see eq. ??»
The conclusions in this paper are interesting. However, this section lacks numerical information. Conclusions should be expanded, estimates should be given in numerical form. For instance, authors demonstrate considerable drop of the ability to generalize the traininfg data for strong covariate shift between training and validation subsets… The conclusions should contain numerical estimates of this drop.
- It would also be useful to give examples of the tcc time series used in the paper.
- Figure 5. It is necessary to give the number of cases in the histograms. Under the picture, in caption, it is useful to indicate which time period the data covers as well as latitude range.
- it is also necessary to expand the technical description of the used models based on neural networks. It is possible to show in figure schematic illustrations of general line of the retrieval algorithms.
-Also I may recomend to authors to pay attention into papers in order to compare the results with obtained data in another studies:
- Kutubuddin Ansari, Tae‐Suk Bae, Jisun Lee Spatiotemporal variability of total cloud cover measured by visual observation stations and their comparison with ERA5 reanalysis over South Korea
- Free M., Sun B., Yoo H.L. Comparison between Total Cloud Cover in Four Reanalysis Products and Cloud Measured by Visual Observations at U.S. Weather Stations/ Journal of Climate. 2016/ 29(6):160115131505009
- Bin Yao, Chao Liu, Yan Yin, Zhiquan Liu, Chunxiang Shi, Hironobu Iwabuchi and Fuzhong Weng Assessment of cloud properties from the reanalysiswith satellite observationsover East Asia (https://amt.copernicus.org/preprints/amt-2019-223/amt-2019-223.pdf)
Reviewer 3 Report
This paper describes an objective method for reporting total all sky camera measurements for total cloud over over the oceans. The method uses a data driven "neural "model derived from shipboard data at various midlatitude and arctic locations.
The introductory material is far too lone taking up ~8 pages out of 18 pages of text. The authors should consider removing section 1.3--cloud classification since it is not relevant to the experiment. The other parts of section 1 could be compressed significantly without loss of information to the reader.
The discussion might include at least a paragraph on how the TCC data can be used for ocean climatology-eg. climate model improvements, and how the improved data will enhance the value of observer visual records.
The paper is only partially complete as noted for example on Line 64 reference? and Line 537 reference missing. The references are not in the proper form for the journal; they lack dates of publication and journal titles, for example.
The english is rough and needs attention, especially for typos and for confusion between singular and plural verbs.
To be acceptable for publication, the authors should go over the ms for errors missing references or information and fix the reference list.
Reviewer 4 Report
This paper proposes a framework for the systematic study of automatic schemes for the retrieval of total cloud cover from all-sky imagery along with the quality measures for the assessment and comparison of the algorithms within the framework. I recommend this paper to be published after minor revision.
(1) I found it is difficult to follow the methodology described in section 3, especially in section 3.3. I’d suggest the authors rewrite this part to make it more clearly and logically.
(2) Table 3: What the quality estimates of PNetOR trained by Region-agnostic training subset look like if validated with the Region-aware validation subsets ?
(3) There are some typos in the manuscript.
Line 64: ‘observations [? ] ’
Line 104: ‘The researchers my consider’, may?
Line 226: ‘type [54? ]’
Line 479: ’will be the following: [ 0, 0, 0, 1, 0, 0, 0, 0, 0] ..’
Line 537:’ accuracy" (Leq 1A , see eq. ?? ).’
Line 595: ‘we in tab. 3 we present’
Round 2
Reviewer 3 Report
The revision I downloaded is poorly edited and proof read. There are problems with indentations and spacing between words. Eq. 5, 9 and 10 are incorrectly written. There are typos and English problems singular vs. plural verbs and omission of prepositions. Errors like Line 40 initial phrase (?). Lines 644-645 is muddled with bold and different case sentence and a run for next major topic. See Line 631 and Table 3, for example--there is an apostrophe in numbers rather than a comma. The references require year of publication of report or journal volume.
If I received the correct revision version from the journal by email, the revision is still rough and not ready for publication. The ms needs considerable editing and proof reading again.
The authors chose not to reduce the length of the introduction. It is too long; it is nearly half of the total pages of the manuscript. I suggest that the section of cloud climatology could be greatly reduced since the material is irrelevant to the main content of the paper (Cloud type by region is not dealt with in the methods described). Another possibility for editorially reducing the introduction by changing the sub- topics as section "2"--context of TCC??
